# Benchmarking Controllers for Low-Cost Agricultural SCARA Manipulators

**DOI:** 10.3390/s25092676

**Published:** 2025-04-23

**Authors:** Vítor Tinoco, Manuel F. Silva, Filipe Neves dos Santos, Raul Morais

**Affiliations:** 1INESC TEC—Institute for Systems and Computer Engineering, Technology and Science, 4200-465 Porto, Portugal; mss@isep.ipp.pt (M.F.S.); filipe.n.santos@inesctec.pt (F.N.d.S.);; 2Department of Engineering, UTAD—University of Trás-os-Montes and Alto Douro, 5000-801 Vila Real, Portugal; 3ISEP/IPP—School of Engineering, Polytechnic Institute of Porto, 4200-072 Porto, Portugal

**Keywords:** sliding mode control, reinforcement learning, PI control, manipulator, agricultural manipulator

## Abstract

Agriculture needs to produce more with fewer resources to satisfy the world’s demands. Labor shortages, especially during harvest seasons, emphasize the need for agricultural automation. However, the high cost of commercially available robotic manipulators, ranging from EUR 3000 to EUR 500,000, is a significant barrier. This research addresses the challenges posed by low-cost manipulators, such as inaccuracy, limited sensor feedback, and dynamic uncertainties. Three control strategies for a low-cost agricultural SCARA manipulator were developed and benchmarked: a Sliding Mode Controller (SMC), a Reinforcement Learning (RL) Controller, and a novel Proportional-Integral (PI) controller with a self-tuning feedforward element (PIFF). The results show the best response time was obtained using the SMC, but with joint movement jitter. The RL controller showed sudden breaks and overshot upon reaching the setpoint. Finally, the PIFF controller showed the smoothest reference tracking but was more susceptible to changes in system dynamics.

## 1. Introduction

The world population is expected to reach 9.7 billion by 2050 [1], which has increased the need for agricultural products. Current labor shortages have motivated the development of agricultural robots to reduce their impact [2,3,4]. Engaging in harvesting demands extensive labor hours and proves physically taxing and repetitive, contributing to prolonged work-related injuries. Moreover, these tasks are typically seasonal, dissuading workers from risking unemployment until the next season [2,3,5,6]. Integrating robots into agriculture addresses labor shortages by assigning repetitive tasks to more adept machines. Nevertheless, introducing robots into agriculture, especially in the context of harvesting, poses several challenges [7,8]. In contrast to an industrial setting, characterized by uniform objects and controlled workspace constraints designed for robot interaction, the agricultural environment is dynamic. Fruits, vegetables, and plants exhibit shape, size, and color variations [5]. Most of these challenges can be resolved given a robotic manipulator with a proper kinematic configuration, sensing capabilities and control algorithms, delivering grasping orientations to swiftly pick up different fruits without damaging them [9].

Currently, commercially available robotic manipulators have prices ranging from EUR 3000 to EUR 500,000 [10], making them too costly to complement the work required during harvesting periods. The main issue with lower-cost manipulators is their inaccuracy, limited sensor feedback, dynamic uncertainties, etc. [11]. A possible way to mitigate the poor performance of these manipulators is through software, such as developing a control architecture that can work in the presence of dynamic uncertainties, friction, and sensor limitations, such as neural network controllers or robust controllers. Various classical and learning-based control methods have been applied to manipulators, but there is a lack of real-world comparisons. This study benchmarked three control strategies: Sliding Mode Control (SMC), Reinforcement Learning (RL), and a novel Proportional-Integral (PI) controller with self-tuning feedforward compensation (PIFF) on a Selective Compliance Assembly Robot Arm (SCARA) manipulator designed for agricultural tasks. Compared to traditional industrial SCARA manipulators, which are typically designed for high precision in controlled environments, the proposed low-cost solution targets agricultural applications, where flexibility and cost-effectiveness are prioritized over extreme precision, making the system more accessible for agriculture, where labor shortages and high equipment costs are significant challenges.

This document is organized in the following way: Section 2 presents literature examples of neural network controllers and sliding mode controllers used in robotic manipulators, followed by robotic manipulators developed for agricultural purposes. Section 3 presents the system design of the proposed manipulator and its kinematic and dynamic configuration. Section 4 presents the developed sliding mode controller, and Section 5 presents the developed neural network controller. Section 6 introduces a novel PI controller with a self-tuning feedforward element. Section 7 presents a comparison between the presented controllers. Finally, a discussion of the three presented controllers and the conclusions to the document are presented in Section 8.

## 2. Related Work

This section presents literature examples of sliding mode controllers, reinforcement learning controllers, and agricultural manipulators.

### 2.1. Classical Control Approaches for Manipulators

Rev.3Classical robotic manipulator control relies on well-established PID-based and model-based controllers. PID controllers are widely used, due to their simplicity and effectiveness in position control [12]. However, they struggle with nonlinearities, disturbances, and unmodeled dynamics, requiring additional compensators. Model Predictive Control (MPC) is another approach that optimizes control inputs over a finite horizon, allowing constraint handling and smoother trajectory tracking [13]. These classical controllers are used in a wide range of manipulator applications and are often combined with other control methodologies. For example, Shojaei et al. [14] developed an observer-based neural adaptive PID controller for robotic manipulators, Londhe et al. [15] developed a PID fuzzy control scheme for underwater vehicle manipulators, Heidar et al. [16] proposed a PID fuzzy controller for a parallel manipulator. Kumar et al. [17] developed a neural network-based PID controller for a one-link manipulator. This controller allowed the use of an unknown system model and could identify system uncertainties that prevailed during the manipulator’s operation. Tang et al. [18] proposed a self-adaptive PID controller based on a radial basis function neural network to resolve the weak adaptive ability and poor robustness of conventional PID controllers.

Classical methods also employ axis control architectures, where high-bandwidth inner-loop controllers regulate motor currents and velocities, while outer-loop controllers handle motion planning and disturbance rejection [19].

### 2.2. Sliding Mode and Neural Network Controllers

Many classic robust controllers, namely Sliding Mode Control, are still prevalent in the literature. SMC is robust to parameter changes, nonlinear disturbances, and uncertainties [20,21]. This controller type drives the system’s state to “slide” on a switching surface. These sliding surfaces represent a desirable and stable system response in the control context. Nadda et al. [22] developed a nonlinear integral SMC for position control of a robotic manipulator with external disturbances, payload variations, and other inherent factors. The controller was tested on a simulated manipulator and benchmarked with a classic Proportional-Integral-Derivative (PID) controller, showing a better tracking performance than the latter. Bhave et al. [23] proposed a third-order sliding mode controller for an underactuated robotic manipulator with no chattering. The controller was tested via simulation, showed robustness to uncertainties, and converged in finite time. Kameyama et al. [24] and Li et al. [25] proposed event-triggered SMCs for reference tracking of robotic manipulators, reducing data communications, as the controller was triggered less often. However, the authors faced problems due to chattering in the control signal and unknown disturbances. Boukadida et al. [26] developed an optimal SMC for trajectory tracking of manipulators by integrating a first-order SMC with a Linear-Quadratic-Regulator (LQR). The controller was tested on a simulated manipulator, and chattering affected the control signal.

Recently, due to advancements in computation, the use of neural network (NN) controllers has risen in popularity in the literature [27,28]. Neural networks can be divided into two subsets, depending on their learning rule: supervised learning, and unsupervised learning [28]. Supervised learning involves algorithms learning patterns from labeled data. Input–output pairs are provided for training, where the algorithm adjusts its parameters to minimize the difference between its predictions and the actual labels. This method is used in various applications, such as image recognition [28,29]. Unsupervised machine learning uses learning patterns from unlabeled data without guidance. Unlike supervised learning, no predefined output labels guide the process. Manipulator NN control mainly uses this method, as joint feedback is unlabeled [29,30].

Nubert et al. [31] used Model Predictive Control (MPC) combined with a NN on a robotic manipulator. The manipulator set point, task space trajectory, and respective control commands were determined using MPC. The NN was trained using offline data from the former and used to approximate the control law. This controller was tested through simulation, and the authors could demonstrate its suitability; however, future work is required to implement it on a real manipulator, and limitations in the input control signals were also reported. Zhou et al. [32] developed a deep reinforcement learning NN for manipulator position control using the Deep Deterministic Policy Gradient (DDPG) algorithm. Their controller was trained through simulation, and only training results were presented. Li et al. [33] used an actor–critic-based reinforcement learning NN for manipulator motion planning. Their controller was also trained through simulation, and only training results were reported. Hu et al. [34] proposed a reinforcement learning NN for controlling a manipulator with dead zone and unknown parameters. This controller was also simulated, but the authors presented realistic results with input constraints, such as maximum joint velocity.

The previously mentioned works used simulations to test and validate the proposed controllers and did not use real manipulators. Using a real environment creates various problems not typically discussed in the literature and that are explored in this document, namely real dynamics, parameter uncertainties, and joint frictions.

### 2.3. Agricultural Manipulators

Robotic manipulation for crop harvesting, particularly tomatoes, has garnered substantial attention in agricultural automation research. In the work by Lili et al. [35], a greenhouse-compatible mobile system was introduced, equipped with a 5-degree-of-freedom (DoF) robotic manipulator. The system utilized a binocular vision setup in combination with Otsu’s thresholding algorithm to distinguish ripe tomatoes from unripe ones. A shear-type end-effector was employed to grip and sever the peduncle of the fruit. The robot navigated its workspace using a configuration space (C-space) strategy for collision-free motion planning. Mohamed et al. [36] developed a seven DoF variable-stiffness manipulator with a soft gripper. The manipulator has agonist–antagonist actuators connected to the joints through flexible tendons. The manipulator uses a color/depth camera to detect the tomato location and an eye-in-hand camera for visual servoing [37]. Oktarina et al. [38] developed a compact four DoF robotic manipulator for tomato harvesting. This manipulator used an integrated eye-in-hand vision system to classify tomatoes by color and determine ripeness. Additionally, an ultrasonic proximity sensor enhanced spatial awareness for improved positioning during picking tasks. Yaguchi et al. [39] introduced a fully autonomous tomato-picking system using a commercially available six DoF UR5 robotic arm and a rotational gripper capable of plucking fruit. A USB stereo vision camera served as the perception module, and the Random Sample Consensus (RANSAC) algorithm was applied to perform sphere fitting for robust fruit localization.

## 3. Manipulator Design

The manipulators previously mentioned feature multiple degrees of freedom, which can result in increased complexity in control algorithms due to the intricacies of their dynamic and kinematic systems. Consequently, this may also lead to higher overall system costs, as the need for additional sensors and actuators arises. Kondo et al. [40] concluded that tomato harvesting only requires horizontal movements, as the peduncle is perpendicular to the floor. Given this, for this work, a three DoF Stackable Selective Compliance Assembly Robot Arm (SCARA) manipulator, shown in Figure 1, is proposed to harvest tomatoes. Rev.3A Proportional-Rotational-Rotational (PRR) configuration was selected due to the confined nature of the workspace (for example, a tomato plant), where dense foliage, including branches and leaves, limits the available space. In such conditions, a smaller tool volume is advantageous, as it improves maneuverability and reduces the risk of damaging the plant. In contrast, an RRP configuration typically results in a larger volume near the manipulator’s tip, which may hinder operation in these restricted environments. Additionally, employing a prismatic joint to lift the manipulator allows the actuator to be positioned at the base, shifting the center of mass closer to the robot’s base. This reduces the torque required by the rotational joints, although it increases the force demand on the prismatic actuator. Despite the simplicity of this kinematic structure, it is subject to the pyramidal effect, where small angular errors and velocities in the proximal joints become amplified as linear errors and velocities at the end-effector. The proposed manipulator will constitute a multi-robotic system with similar manipulators stacked on each other, as shown in Figure 2.

The manipulator structure mainly comprises stainless steel components and weighs 4.20 kg. The first link of the manipulator is an elevator with two shafts that allow the vertical movement of the manipulator in a range of up to 250 mm using a prismatic joint (joint 1) composed of a worm-gear Direct Current (DC) motor and steel wire. The second link of the manipulator has a length of 310 mm and is connected to the first link through a rotational joint (rotational joint 1), and uses the same worm-gear DC motor as joint 1. Finally, the third link has a length of 275 mm and is connected to the second link through a rotational joint, with the same worm-gear DC motor as the previous joints. The stainless steel design allows for a low-cost and robust design, while the worm-gear DC motors allow for a low-cost alternative, with a higher velocity than stepper motors. The mentioned SCARA manipulator is shown in Figure 3.

With this mechanical design, the manipulator presented in Figure 1 has its coordinate frames presented in Figure 4, where $d_{1}$ is the linear displacement of the first joint (prismatic); $\theta_{2}$ and $\theta_{3}$ are the angles of rotation of joint 2 and joint 3, respectively; and $l_{2}$ and $l_{3}$ are the lengths of link 2 and link 3, respectively. Subsequently, the Denavit–Hartenberg (DH) convention was used first to obtain the DH parameters of the manipulator, presented in Table 1, and then to create a transformation matrix between the base and the end-effector, **T**, presented in Equation (Equation 1), with the correct values of $l_{2}$ and $l_{3}$.



(1)
$$T=\left[\begin{array}{cccc} \cos(\theta_{1}+\theta_{2}) & -\sin(\theta_{1}+\theta_{2}) & 0 & l_{2}\cos\left(\theta_{1}\right)+l_{3}\cos\left(\theta_{1}+\theta_{2}\right)\\ \sin(\theta_{1}+\theta_{2}) & \cos(\theta_{1}+\theta_{2}) & 0 & l_{2}\sin\left(\theta_{1}\right)+l_{3}\sin\left(\theta_{1}+\theta_{2}\right)\\ 0 & 0 & 1 & d1\\ 0 & 0 & 0 & 1 \end{array}\right]$$



The forward kinematics of the manipulator, specifically the cartesian coordinates of the manipulator’s tip, are given by the last column of the matrix **T** and presented in Equation (Equation 2).(2)$$\begin{array}{cc} x & =l_{2}\cos\left(\theta_{1}\right)+l_{3}\cos\left(\theta_{1}+\theta_{2}\right)\\ y & =l_{2}\sin\left(\theta_{1}\right)+l_{3}\sin\left(\theta_{1}+\theta_{2}\right)\\ z & =d_{1} \end{array}$$

The inverse kinematics are the inverse kinematics for a two DoF Rotational-Rotational (RR) manipulator, with both joints rotating on the same plane, as shown in Equation (Equation 3).(3)$$\begin{array}{cc} d_{1} & =z\\ \theta_{2} & ={\mathrm{atan}}_{2}\left(y,x\right)-{\mathrm{atan}}_{2}\left(l_{3}\sin\left(\theta_{3}\right),l_{2}+l_{3}\cos\left(\theta_{3}\right)\right)\\ \theta_{3} & =\mathrm{acos}(\frac{x^{2}+y^{2}-l_{2}^{2}-l_{3}^{2}}{2l_{2}l_{3}}) \end{array}$$

The previous equation can have two possible solutions for the kinematic configuration. To remove this ambiguity and have only one possible solution, $\theta_{3}$ can be defined as Equation (Equation 4), where the sign of $sin(\theta_{3})$ will define which one of the two possible solutions will be used.(4)$$\begin{array}{cc} \; & cos\left(\theta_{3}\right)=\frac{x^{2}+y^{2}-l_{2}^{2}-l_{3}^{2}}{2l_{2}l_{3}}\\ \; & \sin\left(\theta_{3}\right)=\pm \sqrt{1-\cos{\left(\theta_{3}\right)}^{2}}\\ \; & \theta_{3}={\mathrm{atan}}_{2}\left(\pm \sqrt{1-\cos{\left(\theta_{3}\right)}^{2}},\cos\left(\theta_{3}\right)\right) \end{array}$$

The general equation for the manipulator’s inverse dynamics is shown in Equation (Equation 5) [41], where $\tau \in R^{3}$ is the actuator torque vector; **M** $\in R^{3,3}$ is the mass and inertia matrix; **C** $\in R^{3}$ is the Coriolis and centripetal vector; **G** $\in R^{3}$ is the gravity vector; and (q, $\dot{q}$, $\ddot{q}$) $\in R^{3}$ are the joint position, velocity, and acceleration vectors, respectively.(5)$$\tau =M\left(q\right)\ddot{q}+C\left(q,\dot{q}\right)+G\left(q\right)$$

The inertia and mass matrix M is shown in Equation (Equation 6) and each element of M is presented in Equation (Equation 7), where $l_{ncm}$ is the length to the center of mass of link *n*, $m_{n}$ is the mass of link *n*, $m_{n,n+1}$ is the mass of the joint between link *n* and link n+1, and *r* is the radius of the pulley pulling the prismatic joint ($q_{1}$) of the manipulator. The manipulator configurations for the prismatic joint and for the rotational joints are shown in Figure 5a and Figure 5b, respectively.(6)$$M=\left[\begin{array}{ccc} M_{11} & M_{12} & M_{13}\\ M_{21} & M_{22} & M_{23}\\ M_{31} & M_{32} & M_{33} \end{array}\right]$$(7)$$\begin{array}{cc} M_{11} & =m_{1}r\\ M_{12} & =M_{13}=M_{21}=M_{31}=0\\ M_{22} & =l_{2}^{2}m_{23}+l_{2\mathrm{cm}}^{2}m_{2}\\ \; & +m_{34}(l_{2}^{2}+2\cos\left(q_{3}l_{2}l_{3}+l_{3}^{2})\right)\\ & +m_{3}(l_{2}^{2}+2\cos\left(q_{3}l_{2}l_{3\mathrm{cm}}+l_{3\mathrm{cm}}^{2})\right)\\ M_{23} & =m_{34}(l_{3}^{2}+l_{2}\cos\left(q_{3}l_{3})\right)\\ & +m_{3}(l_{3\mathrm{cm}}^{2}+l_{2}\cos\left(q_{3}l_{3\mathrm{cm}})\right)\\ M_{32} & =m_{34}(l_{3}^{2}+l_{2}\cos\left(q_{3}l_{3})\right)\\ & +m_{3}(l_{3\mathrm{cm}}^{2}+l_{2}\cos\left(q_{3}l_{3\mathrm{cm}})\right)\\ M_{33} & =m_{34}l_{3}^{2}+m_{3}l_{3\mathrm{cm}}^{2} \end{array}$$

The Coriolis and centripetal force vector is shown in Equation (Equation 8), and each element of the vector is presented in (Equation 9)(8)$$C=\left[\begin{array}{c} C_{11}\\ C_{21}\\ C_{31} \end{array}\right]$$(9)$$\begin{array}{cc} C_{11} & =0\\ C_{21} & =-l_{2}\sin\left(q_{3}\right)\left(\dot{q_{3}}\right)\\ & \left(l_{3}m_{34}\dot{q_{3}}+2l_{3cm}m_{3}\dot{q_{2}}+l_{3cm}m_{3}\dot{q_{3}}\right)\\ C_{31} & =l_{2}\sin\left(q_{3}\right)\left(l_{3}m_{34}+l_{3cm}m_{3}\right){\dot{q_{2}}}^{2} \end{array}$$

The gravity vector G is shown in Equation (Equation 10), where *g* is the gravitational acceleration constant.(10)$$G=\left[\begin{array}{c} m_{1}rg\\ 0\\ 0 \end{array}\right]$$

Joints $q_{2}$ and $q_{3}$ perform the planar movement on the *xy* plane, and gravity does not affect their movement. Furthermore, the position of the prismatic joint does not influence the gravity vector, only its mass and the radius of the pulley.

## 4. Sliding Mode Control

Sliding mode control was initially developed for controlling systems with uncertain dynamics and external disturbances and is based on the concept of “sliding surfaces”, which involves creating a dynamic system that exhibits a specific behavior when transitioning between different states or modes. These sliding modes represent a desirable and stable system response in the control context. The principle of SMC is to drive the system’s state onto a designated “sliding surface” and maintain it there to achieve desired control objectives.

### 4.1. Sliding Mode Control Model

Considering the control of the presented manipulator, the state **X** = [**x**, $\dot{x}$] represents the position, **x**, and velocity, $\dot{x}$, of each joint. For control purposes, the sliding surface hyperplane is designed in the error state space, with the *x* axis being the position error, **e**, and the *y* axis being the velocity error, $\dot{e}$, as shown in Equation (Equation 11), where ($x_{d}$, $\dot{x_{d}}$) $\in R^{3}$ are the desired joint positions and desired joint velocities, respectively.(11)$$\begin{array}{cc} e & =x-x_{d}\\ \dot{e} & =\dot{x}-\dot{x_{d}} \end{array}$$

The sliding surface, **S**, is first defined as Equation (Equation 12), where λ is the sliding surface gain that represents a proportionality factor that scales the error term to influence the behavior of the sliding mode controller, determining the rate at which the system approaches the sliding surface.(12)$$S=\dot{e}+\lambda e$$

To create a hyperplane, the sliding surface is equalled to zero, and thus Equation (Equation 12) can be rewritten as Equation (Equation 13), where λ is now the slope of the hyperplane, shown in Figure 6.(13)$$\dot{e}=-\lambda e$$

The chosen control law is defined using the system’s dynamic model, with both the desired acceleration and a switching function dependent on the signal of the sliding surface, as shown in Equation (Equation 14) [42,43], where $\tau \in R^{3}$ is the actuator torque vector; **M** $\in R^{3,3}$ is the mass and inertia matrix; **C** $\in R^{3}$ is the Coriolis and centripetal vector; **G** $\in R^{3}$ is the gravity vector; and (q, $\dot{q}$, and $\ddot{q}$) $\in R^{3}$ are the joint position, velocity, and acceleration vectors, respectively. where $\ddot{q_{d}}\;\in \;R^{3}$ is the desired joint acceleration vector, K is the switching gain diagonal matrix, and sign(S) returns the sign of the sliding surface.(14)$$\tau =M\left(q\right)\left(\ddot{q}+K\mathrm{sign}\left(S\right)\right)+C\left(q,\dot{q}\right)+G\left(q\right)$$

The DC motor on each joint receives a Pulse Width Modulation (PWM) signal. To generate this signal, the calculated torque is converted to voltage and then to PWM duty cycle using Equation (Equation 15), where $K_{i}$ is the motor torque constant, *N* is the gearbox ratio, $R_{m}$ is the motor electrical resistance, and *U* is the motor nominal voltage.(15)$$\begin{array}{c} V=\frac{\tau }{K_{i}}\frac{1}{N}R_{m}\\ DC\%=\frac{V}{U}100 \end{array}$$

### 4.2. Experimental Results

The controller was first simulated using MATLAB (https://www.mathworks.com/, accessed on 30 March 2025). The simulated manipulator was modeled to respond similarly to the real manipulator. Furthermore, Gaussian noise was added to the joint position feedback to simulate a noisy sensor, and a motor dead zone of 15 % of the motor nominal voltage was added. Moreover, to simulate a proper motor control system, the voltage was set to 0 V for 10 ms whenever the control signal changed sign, to protect the H-bridge. The switching gain K and the sliding surface gain λ were chosen using trial-and-error and considering the trade-off between system stability and response time in the simulated environment. The experiment consisted of driving the joints to two different reference points. The reference tracking of the simulated manipulator using SMC is shown in Figure 7.

The SMC changes the sign of the control signal depending on the current error state space. This causes high-frequency chattering, as shown in Figure 8, and can potentially damage the hardware.

To mitigate the chattering effect, the sign(S) element in Equation (Equation 14) is replaced by a saturation function satr(S,ϕ) and the latter equation can be rewritten into Equation (Equation 16) [42], where $\phi \in R^{3}$ is the vector of elements $\phi_{n}$ that define the saturation boundaries, $S_{n}$ is the sliding surface of each joint, and *n* is the joint number.(16)$$\begin{array}{cc} \; & \tau =M\left(q\right)\left(\ddot{q}+K\mathrm{satr}\left(S,\phi \right)\right)+C\left(q,\dot{q}\right)+G\left(q\right)\\ \; & \mathrm{satr}\left(S_{n},\phi_{n}\right)=\left\{\begin{array}{cc} \mathrm{sign}(S_{n}) & |S_{n}|\geq \phi_{n}\\ \frac{S_{n}}{\phi_{n}} & |S_{n}|<\phi_{n} \end{array}\right. \end{array}$$

The saturation function creates a line with slope $\frac{S_{n}}{\phi_{n}}$ to the saturation value $\pm \phi_{n}$, and the controller will not change the control sign instantly, making it smoother. The reference tracking using the SMC with the saturation function and the respective control signal are shown in Figure 9 and Figure 10, respectively.

Both the reference tracking and the control signal of each joint were relatively smoother than without the saturation function. However, the control response was also relatively faster but generated overshoots on the joint with higher moments of inertia. Nevertheless, the saturation function is necessary not to damage the manipulator hardware, and the controller will use a saturation function when deploying.

The same reference points were applied to the real manipulator, as shown previously in Figure 3. The controller was ported from the MATLAB environment into the RP2040 (https://www.raspberrypi.com/products/rp2040/, accessed on 30 March 2025) 32-bit microcontroller present on the embedded boards that control each joint [44]. The reference points were generated in the MATLAB environment and sent to a Robot Operating System 2 (ROS2) to be sent to the microcontroller. The reference tracking for the real manipulator with three sets of loads is shown in Figure 11, and the control signals corresponding to no-load are shown in Figure 12. Furthermore, the error of the joint movements with all loads is present in Figure 13.

The SMC successfully controlled each joint with the reference point on the real manipulator with no load, 500 g, and 1 kg. However, delays between the ROS2 network and the microcontroller may have slightly affected the controller’s performance. Furthermore, the motor dead zone of the real manipulator was more noticeable than the simulated manipulator, and thus, the manipulator would stutter during movement and have steady-state error. Increasing the controller gain on each joint caused instability, namely on rotational joint 2, and thus, the control signal did not reach high values. Nevertheless, the controller enabled reference tracking, without creating a chattering effect on the control signal. Table 2 shows the Root Mean Square (RMS) error of the joint’s trajectory tracking for all loads. The slow start on the prismatic joint caused the RMS error to range from 9.0 cm to 9.2 cm, and could be significantly lower. The slow response from the rotational joint 2 also caused a relatively high RMS error. Even though the reference value is higher for the rotational joint 2 than for the rotational joint 1, the RMS error of 1.388 rad to 1.447 rad could be lower with more adequate SMC parameters.

## 5. Reinforcement Learning

Reinforcement Learning (RL) is a machine learning paradigm in which an agent learns to make decisions by interacting with an environment and receiving rewards or penalties based on its actions. The agent continuously learns through trial and error, exploring different actions and receiving feedback in the form of rewards, striving to maximize the cumulative reward over time [45,46]. An RL system typically consists of two key components: the actor, which selects actions based on the current policy, and the critic, which evaluates the chosen actions by estimating their value and helping the actor improve its policy.

### 5.1. Reinforcement Learning Control Model

A deep deterministic policy gradient RL algorithm was used to control each manipulator joint. The DDPG algorithm is based on an actor–critic network. It combines deep learning and policy gradient methods to learn complex tasks in environments where actions are continuous [45,46]. Figure 14 shows a diagram of the actor–critic network.

In the context of the presented manipulator, the actions are voltages in the form of a PWM signal. In the actor–critic architecture, the actor network is responsible for learning the optimal policy, $\pi^{*}$, that generates the highest reward, which, depending on the state, determines the action. The critic network evaluates the actions generated by the actor network and determines how good or bad the action is in a given state. For this work, a reinforcement learning agent was created for each joint, to enable individual joint control. The agent creation, training, and deployment were performed using the MATLAB Reinforcement Learning toolbox (https://www.mathworks.com/products/reinforcement-learning.html, accessed on 30 March 2025). The optimal policy can be defined by Equation (Equation 17), where E is the expected cumulative reward, γ is the discount factor, *r* is the reward, and *t* is the time-step.(17)$$\pi^{*}=\arg\max_{\pi }E\left[\sum_{t=0}^{\infty }\gamma^{t}r_{t}\right]$$

Reinforcement learning agents usually utilize functions to assist in decision-making. The state-value function (V(s)), shown in Equation (Equation 18), estimates the expected cumulative reward achievable from a given state *s*, and the action-value function (Q(s,a)), shown in Equation (Equation 19), estimates the expected cumulative reward achievable from action *a* in state *s*. The actor is responsible for selecting actions based on the current state. Simultaneously, the critic evaluates the chosen actions by estimating the value of the action taken and updating the actor’s policy accordingly.(18)$$V\left(s\right)=E[r_{t}+\gamma V\left(s_{t+1}\right)|s_{t}=s]$$(19)$$Q\left(s,a\right)=E[r_{t}+\gamma Q\left(s_{t+1},a_{t+1}\right)|s_{t}=s,a_{t}=a]$$

The DDPG algorithm aims to improve the previously shown value functions so that they converge towards the optimal policy.

Three separate agents were created to control each joint individually. The created agents observe the state $s=[\omega_{n}$ $\dot{\omega_{n}}$ $e_{n}$ $e_{n-1}$ $u_{n-1}{]}^{T}$, where $\omega_{n}$ is the current joint position, $\dot{\omega_{n}}$ is the current joint velocity, $e_{n}$ is the current error, $e_{n-1}$ is the previous error, and $u_{n-1}$ is the previous action. Since each agent controls a joint, the action space comprises a single voltage value a=V per agent.

To increase the cumulative reward of each iteration, the reward function present in Equation (Equation 20) was used. The presented reward function rewards low positioning errors and penalizes high velocities when the error is low. This trains the agent to reduce the joint’s angular velocity as it reaches the desired position.(20)$$\begin{array}{ccc} \; & r_{1}=-1.5\times \left|e_{n}\right|\\ \; & r_{2}=-0.5 & |e_{n}\left|<0.25\;\mathrm{rad}\wedge \right|\dot{w_{n}}|>2.5\;\mathrm{rad}\;s^{-1}\\ \; & r_{3}=1.0 & |e_{n}|\leq 0.01\;\mathrm{rad}\\ \; & r=r_{1}+r_{2}+r_{3} \end{array}$$

Contrary to more classical control methods, the trained agent could not be altered or adjusted to increase its performance on the real manipulator, such as in a PID controller where the gains can be manually altered. Given this, the reinforcement learning training was performed in two parts: (*i*) simulation training, and (*ii*) real scenario training using the RL Agent block present in the MATLAB Reinforcement Learning toolbox. The hyperparameters used were obtained through trial and error, and are shown in Table 3, where the learning rate, as the name implies, defines how fast the algorithm “learns”, and the discount factor represents the proportion of future and current rewards. The value γ=0.90 sets future rewards as more important than the current rewards. The batch size defines the number of samples per iteration, and the sample time defines the period between sensing and acting on the environment.

### 5.2. Experimental Results

The agent was trained using the same environment as the sliding mode controller and with the same motor dead zone of 15%. Furthermore, the same reference was used in the reinforcement learning simulation as the one used with the sliding mode controller. Figure 15 shows the simulated reference tracking using reinforcement learning. These results showed a fast convergence rate for the rotational joints. However, the prismatic joint showed a slower convergence. Nevertheless, the simulated manipulator was able to reach the reference smoothly.

The trained agent was deployed on the real manipulator using the MATLAB ROS (https://www.mathworks.com/products/ros.html, accessed on 30 March 2025) toolbox to send PWM signals to the joints. Figure 16 shows the trajectory tracking of the joints without further training. The differences between the real and simulated environments are clearly visible in the figure shown, as tracking instability was observed.

The agent used was subjected to training in the real environment. The results of the trajectory tracking with the trained agent and different loads are shown in Figure 17. Furthermore, Figure 18 shows the tracking error of each joint for all loads. The first joint could not reach the upper reference after 300 episodes. However, the rotational joints could follow the trajectory with low error for all loads.

The reinforcement learning controller showed promising results, as it was able to follow the given reference points; however, the prismatic joint struggled to reach the reference point but did not show any instability. Unlike other control methodologies, neural network controllers cannot be adjusted or changed to minimize specific errors and require a training process to solve the errors. The training process can take a long period to perform and does not guarantee correct performance of the controllers. Nevertheless, as mentioned before, the controller showed promising results, and with more invested research, it could be deployed on real scenarios outside of simulated and laboratory environments. Table 4 shows the RMS error of all joint movements for different loads. The rotational joint 1 showed an RMS error of 7.5 cm to 9.1 cm, which cannot be mitigated by tuning parameters and only by re-training the network. The other joints also showed a slight difference in RMS between different loads; however, considering the change in the reference, the RMS error in radians for both rotationals was significantly lower in angular terms than the RMS error in meters for the prismatic joint.

## 6. PI Control with Self-Tuning Feedforward Element

A self-tuning feedforward element was developed to improve the PI controller’s response to motor stalls and dead zones. The controller was developed for the rotational joints of the manipulator presented in this work, whilst the prismatic joint used a simple P controller. The reason for using this type of P controller is that this joint moves slowly, with a maximum velocity of 0.15 m s^−1^ for a duty cycle of 100%.

Each rotational joint has a cascaded controller that comprises a P position controller that feeds into a PI controller with a novel self-tuning feedforward (PIFF). The diagram for this controller is presented in Figure 19.

The self-tuning feedforward block serves as a “universal” block for any of the rotational joints of the manipulator. With this, the main PI controller will have the same *K_p_* and *K_i_* gains for all rotational joints of all manipulators, without needing to be calibrated. For the purposes of this work, *K_p_* = 10 and *K_i_* = 5 were used. This block works with the PI controller’s integral part to increase the PWM’s duty cycle until the desired movement has been achieved. It does this by comparing the current velocity with the target velocity and incrementing or decrementing the feedforward value, much like the integrative element, as shown in the diagram in Figure 20. The difference between this increment and the integral part of the controller is that this increment only happens at half the controller’s frequency, and the feedforward value is saved in memory.

At a 10 Hz frequency, each time the feedforward is updated, its value is saved in a 2D matrix in the microcontroller’s flash memory. The matrix indexes are related to the target velocity, angular position, and rotation direction. The goal is to have a predetermined feedforward value for each angular position, direction, and target velocity. The reason for this is that, during long-term operations, the joints will experience increased friction and dynamics at certain angular positions and directions, due to the manipulator’s components wearing down. This will require future calibrations of the controller gains, and this process might not be feasible. The self-tuning feedforward block is expected to compensate for these changes in dynamics and friction.

If the velocity reference is null, then the feedforward will always be 0; however, if it is not null, then the controller will obtain the feedforward value from memory given the velocity reference, angular position, and direction, as shown in the diagram in Figure 21.

The way the program correlates the target velocity, position, and direction for the feedforward table is shown in Algorithms 1 and 2.

The controller is designed to manage target velocities within the range of −3 rad s^−1^ to 3 rad s^−1^. The sign of the velocity indicates its direction. The program rounds each target velocity to the nearest integer or the nearest half-integer. For instance, a velocity of 3.3 rad s^−1^ rounds to 3.5 rad s^−1^, and a velocity of 4.8 rad s^−1^ rounds to 5 rad s^−1^. This rounding ensures that the target velocity increments by 0.5 rad s^−1^, resulting in 13 possible index values, from −3 rad s^−1^ to 3 rad s^−1^. Algorithm 1 converts the rounded velocity into a matrix index.

For the position correlation, shown in Algorithm 2, the angular position will be confined to −2.8 rad and 2.8 rad. The position is then converted from radians to degrees and divided by 10, having an integer as output. This separates the indexes by increments of 10^*o*^. The third joint has more angular positions than the second one, so the motor number determines the number of angular positions.
**Algorithm 1:** Convert velRef to Table Index
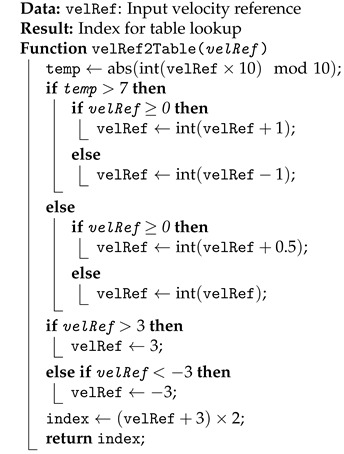


**Algorithm 2:** Convert angle_ to Table Index

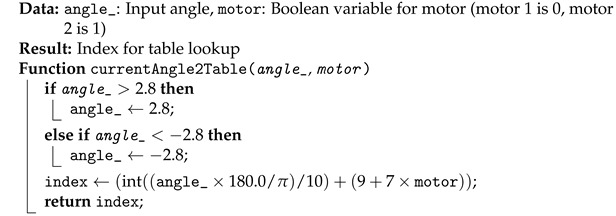



The reference tracking results of the developed PIFF controller are shown in Figure 22, and the respective tracking error is shown in Figure 23.

The prismatic joint used a generic P controller and responded better than the previously presented RF and SMC results. The rotational joints used the PIFF controller. The first rotational joint was very susceptible to differences in the dynamics, due to the load change, specifically with the 1 kg load. However, the second rotational joint showed better results, as there was no substantial difference between the different loads. The reason for this is that the first rotational joint is further back from the manipulator’s tip and has more inertia than the second rotational joint. Table 5 shows the RMS value for each movement of all joints. The RMS error of the rotational joint 1 with the 1 kg load was 0.179 rad to 0.159 rad, due to the higher dynamics caused by the higher load. The rotational joint 2 had a higher difference in reference values (−2.1 rad to 2.1 rad compared to the −1.2 rad to 1.2 rad of the rotational joint 1), and thus had a higher RMS error value of 1.378 rad to 1.465 rad. However, the difference in the RMS error between loads was lower than for rotational joint 2.

## 7. Discussion

The presented controllers were able to perform trajectory tracking to reach the target reference in a reasonable time. Although the general performance of the controllers was suitable for reference tracking, some issues were present. The sliding mode controller showed the best results; however, the manipulator jittered frequently and did not present smooth joint movement. The reinforcement learning controller did not jitter at all, but did not reach the reference on the prismatic joint. Furthermore, the controller performed student stops when reaching the reference at a high velocity, which caused some overshoots. The PI controller with a self-tuning feedforward element had a smoother response. The problem with this controller was that, given the reduced error as the joint approached the reference, the generated reference velocity decreased, and the motor reached the dead zone. The self-tuning feedforward element resolved this issue, but the joint slowed drastically before reaching the reference. Furthermore, the controller was more susceptible to load changes than the previous controllers, as the first rotational joint responded much slower to the 1 kg load.

## 8. Conclusions

This paper presented the implementation and performance comparison of three distinct controllers—a Sliding Mode Controller, a Reinforcement Learning Controller, and a novel PI controller with a Self-Tuning Feedforward Element—implemented on a low-cost agricultural manipulator.

The presented controllers were able to perform trajectory tracking to reach the target reference in a reasonable time. Although the general performance of the controllers was suitable for reference tracking, some issues were present. The sliding mode controller exhibited a rapid response and robustness to disturbances, achieving low tracking errors across varied loads. However, the significant jitter observed in joint movements poses concerns about mechanical wear and potential crop damage. This aligns with existing literature that acknowledges SMC’s robustness but highlights challenges related to chattering effects [47], which limits its use in tasks requiring fine control and precision, such as delicate harvesting tasks. While it provides quick feedback in dynamic conditions, its lack of smoothness is a disadvantage in tasks where accuracy is critical. The reinforcement learning controller demonstrated adaptability without manual tuning, but faced limitations in adequately reaching the designated setpoints, especially at higher velocities. These observations are consistent with studies emphasizing the potential of data-efficient RL in learning control policies for low-cost manipulators, while also noting the necessity for extensive training to achieve desired performance levels [11]. Furthermore, the computational resources and time required for training may be a barrier, particularly in applications that demand immediate responses to changes, such as fruit harvesting in dynamic environments. The PI controller with a self-tuning feedforward element offered smoother motion trajectories, effectively compensating for varying friction and dynamic conditions. However, its higher sensitivity to payload variations, particularly affecting rotational joints, could impact performance consistency. This observation aligns with findings from studies on advanced control strategies, which suggest that while such controllers can enhance precision, they may require careful tuning to effectively handle varying payloads.

In the robotic manipulator control domain, several other advanced methodologies have been explored [48]. Adaptive Control adjusts parameters in real time to cope with system uncertainties and external disturbances, enhancing performance in dynamic environments. Model Predictive Control (MPC) utilizes system models to predict and optimize future behavior, offering precise trajectory tracking and robustness to disturbances. Fuzzy Logic Control (FLC) provides robustness and adaptability, which is particularly beneficial in unstructured environments, by handling system uncertainties and nonlinearities.

While SMC, RL, and PIFF controllers each present unique advantages, integrating features from state-of-the-art control strategies—such as the adaptability of adaptive control, the predictive capabilities of MPC, or the robustness of fuzzy logic—could lead to more effective control solutions for low-cost agricultural SCARA manipulators. Future research should focus on hybrid approaches that combine these strengths to address the specific challenges of agricultural automation.

## Figures and Tables

**Figure 1 sensors-25-02676-f001:**
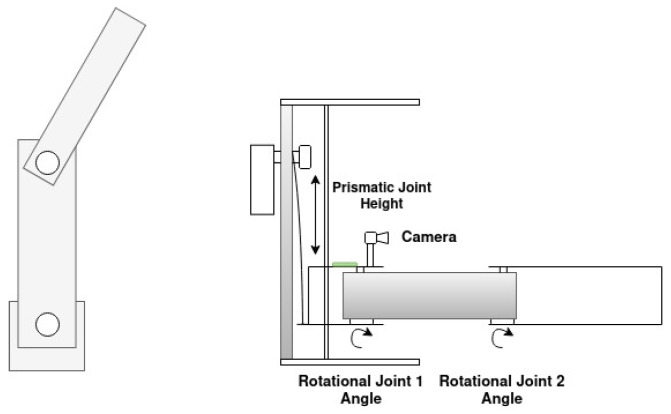
SCARA manipulator diagram: top view (**Left**) and side view (**right**).

**Figure 2 sensors-25-02676-f002:**
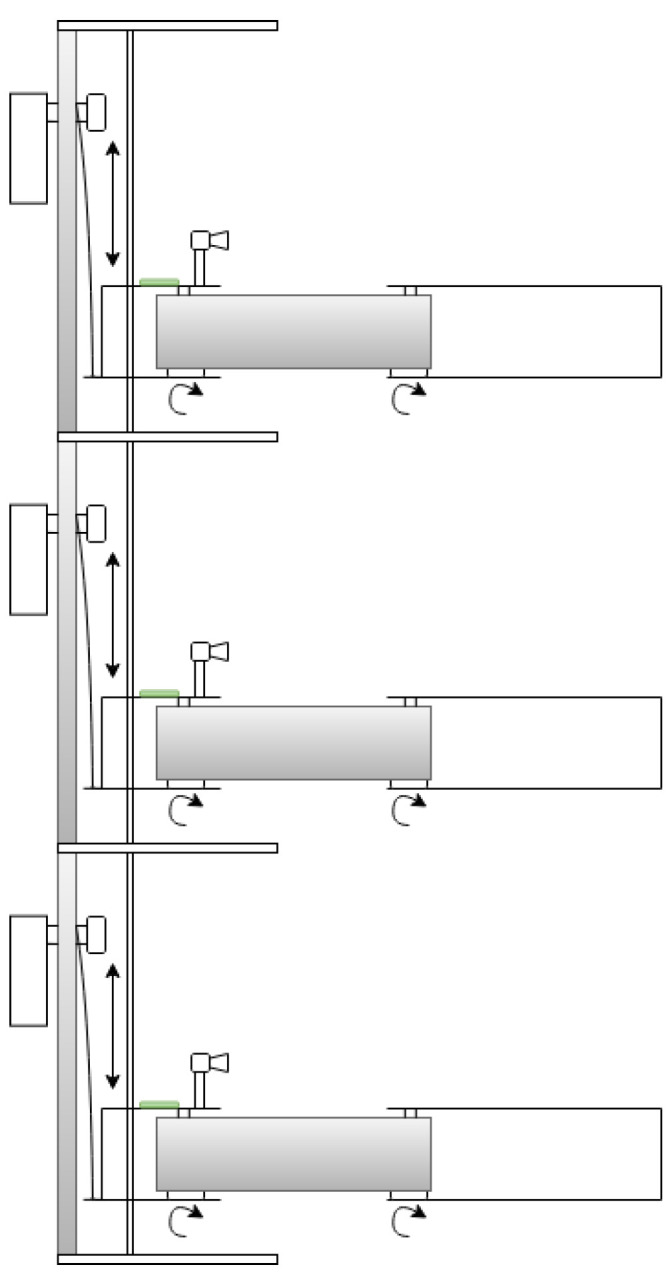
Multi-manipulator concept.

**Figure 3 sensors-25-02676-f003:**
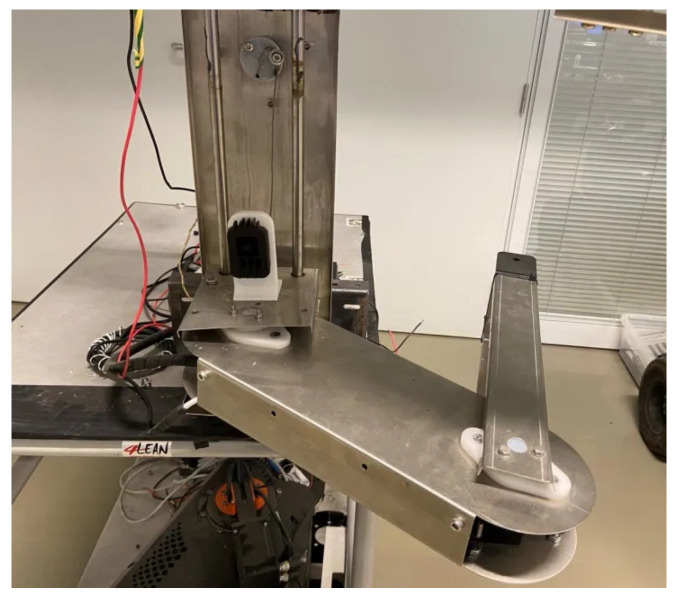
Proposed SCARA manipulator.

**Figure 4 sensors-25-02676-f004:**
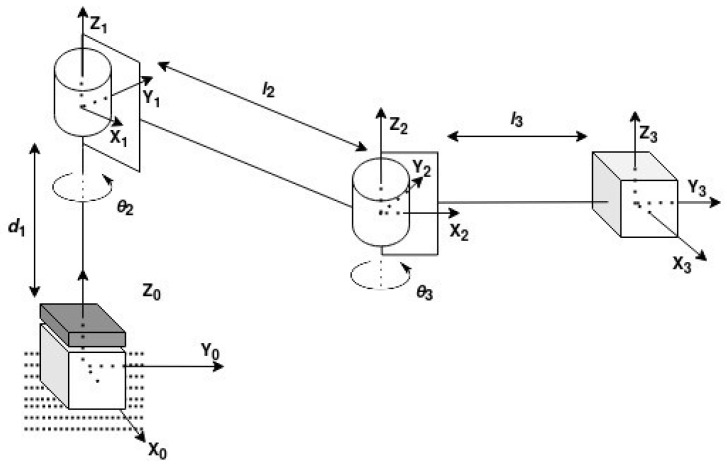
Manipulator coordinate frames.

**Figure 5 sensors-25-02676-f005:**
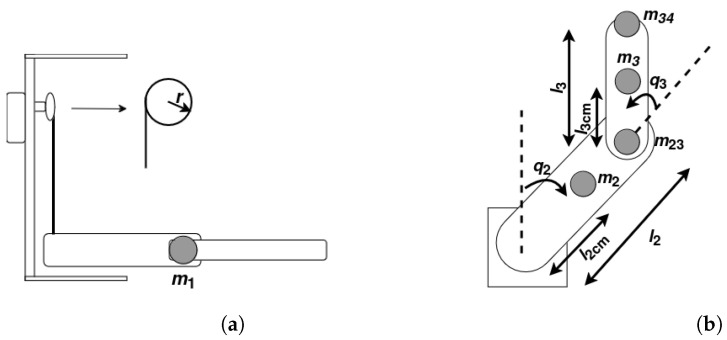
Joint configurations.(**a**) prismatic joint configuration. (**b**) rotational joints configuration.

**Figure 6 sensors-25-02676-f006:**
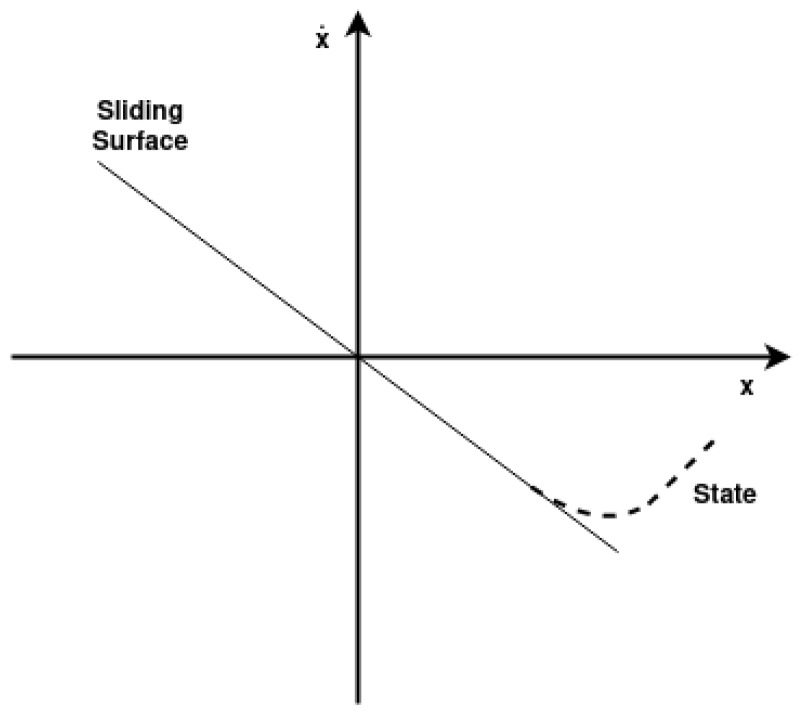
Sliding surface hyperplane.

**Figure 7 sensors-25-02676-f007:**
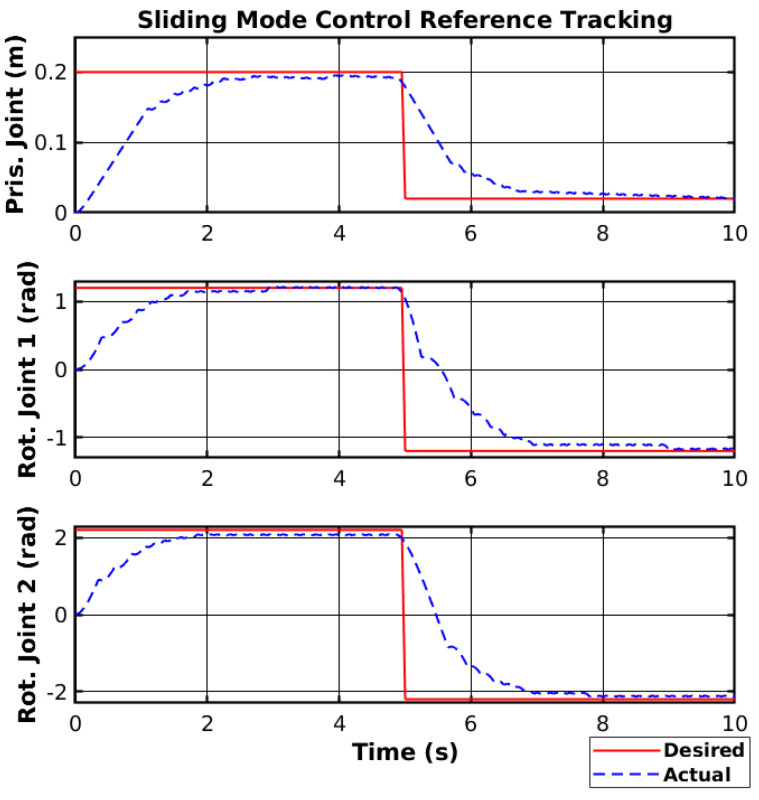
Simulated SMC reference tracking.

**Figure 8 sensors-25-02676-f008:**
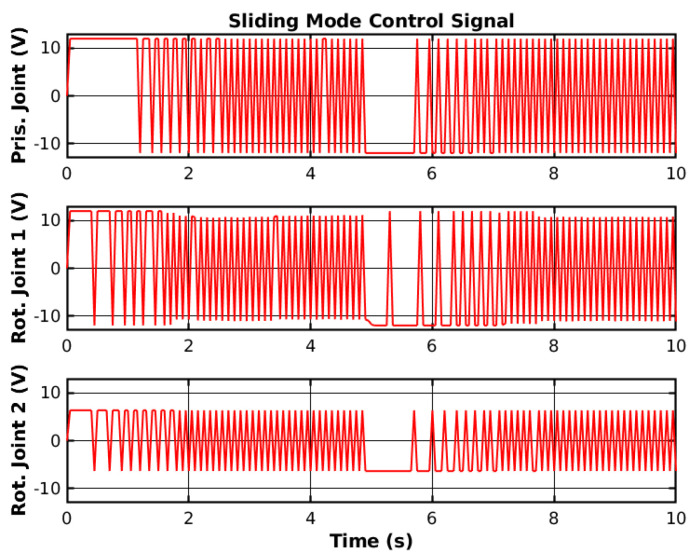
Simulated SMC control signal.

**Figure 9 sensors-25-02676-f009:**
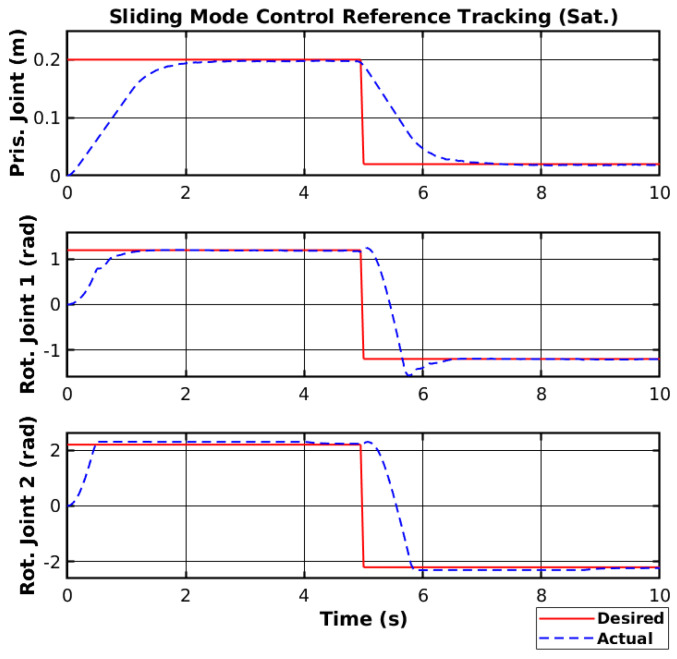
Simulated SMC reference tracking (Sat.).

**Figure 10 sensors-25-02676-f010:**
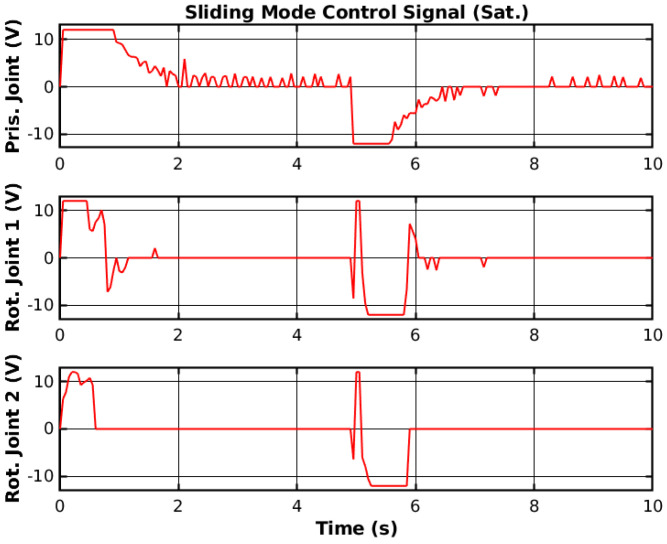
Simulated SMC control signal (Sat.).

**Figure 11 sensors-25-02676-f011:**
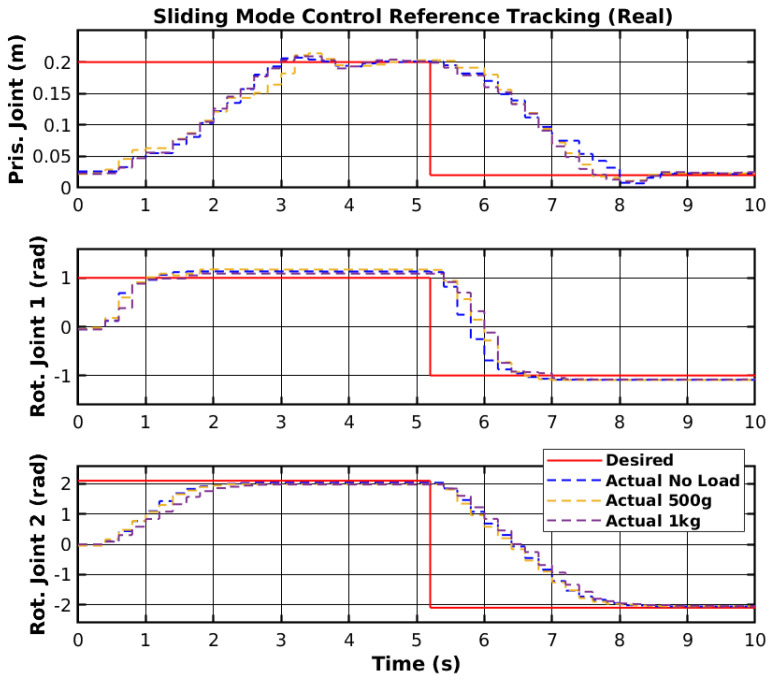
SMC reference tracking.

**Figure 12 sensors-25-02676-f012:**
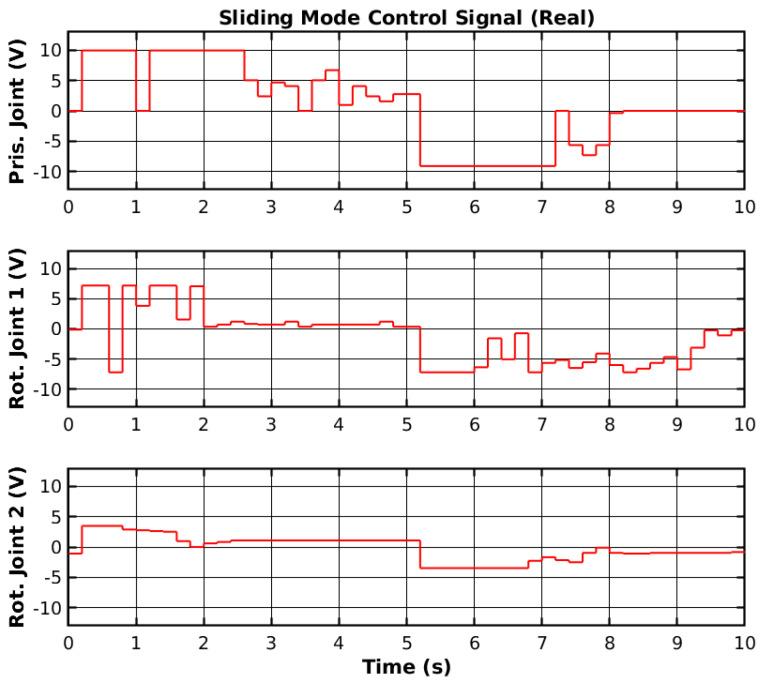
SMC control signal.

**Figure 13 sensors-25-02676-f013:**
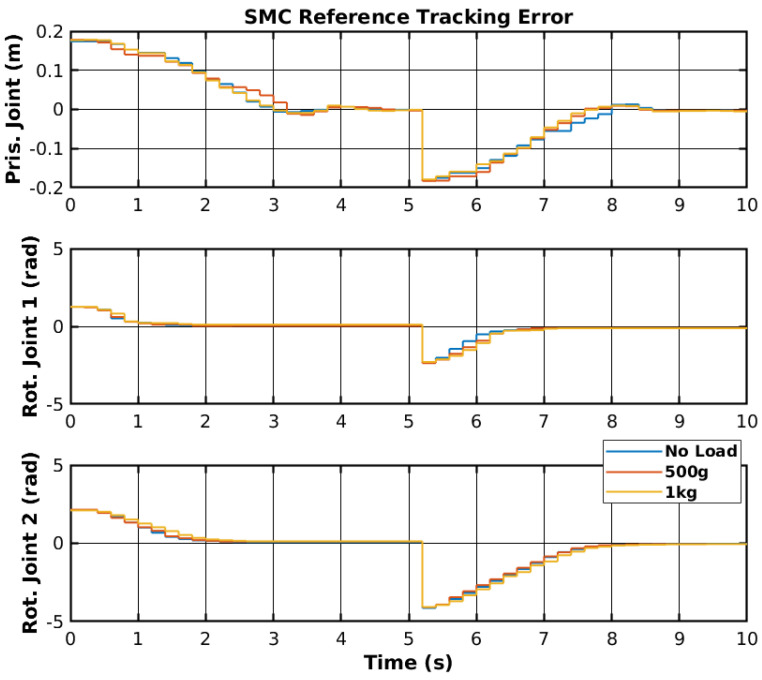
SMC reference tracking error.

**Figure 14 sensors-25-02676-f014:**
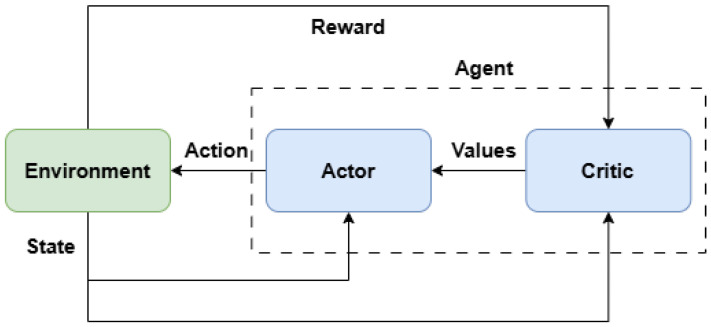
Actor–critic network generic diagram.

**Figure 15 sensors-25-02676-f015:**
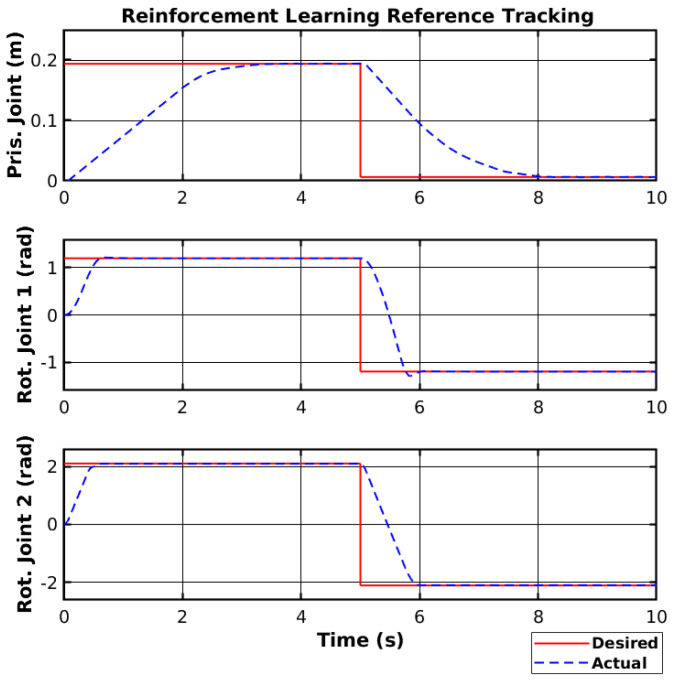
Reinforcement learning reference tracking.

**Figure 16 sensors-25-02676-f016:**
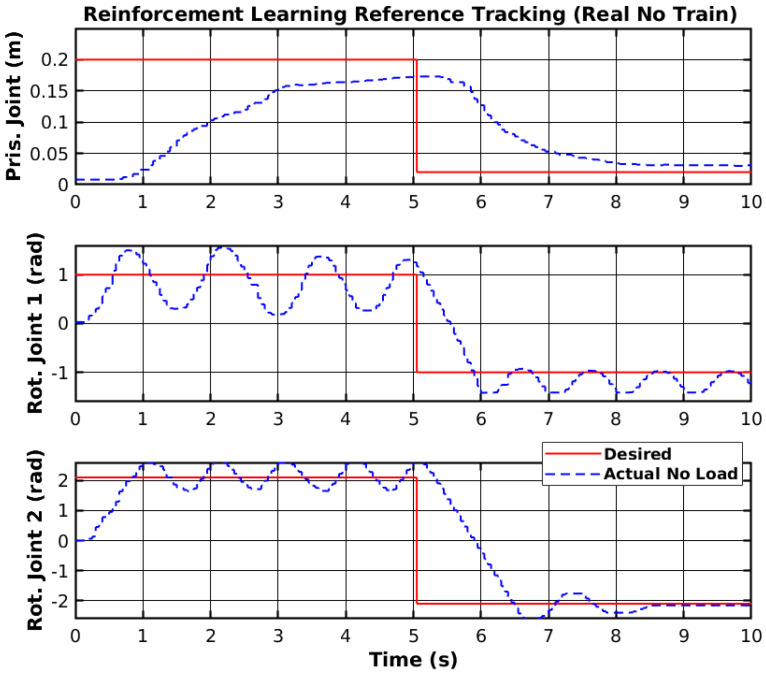
Reinforcement learning reference tracking (real no train).

**Figure 17 sensors-25-02676-f017:**
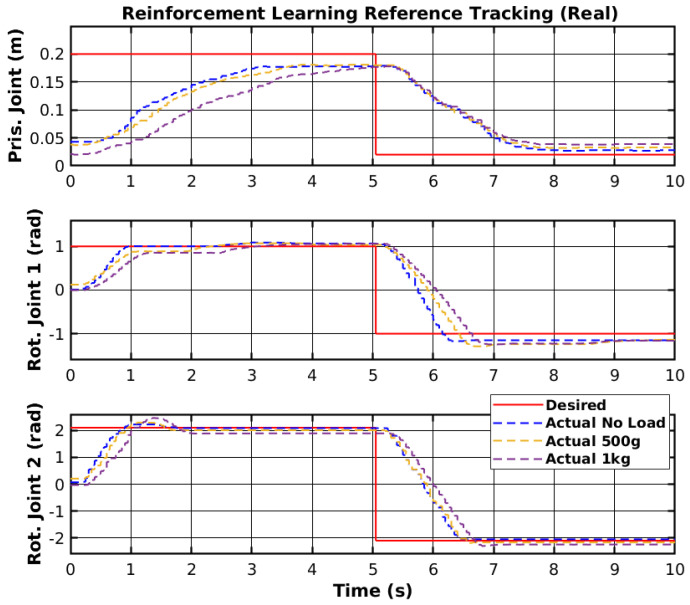
Reinforcement learning reference tracking (real).

**Figure 18 sensors-25-02676-f018:**
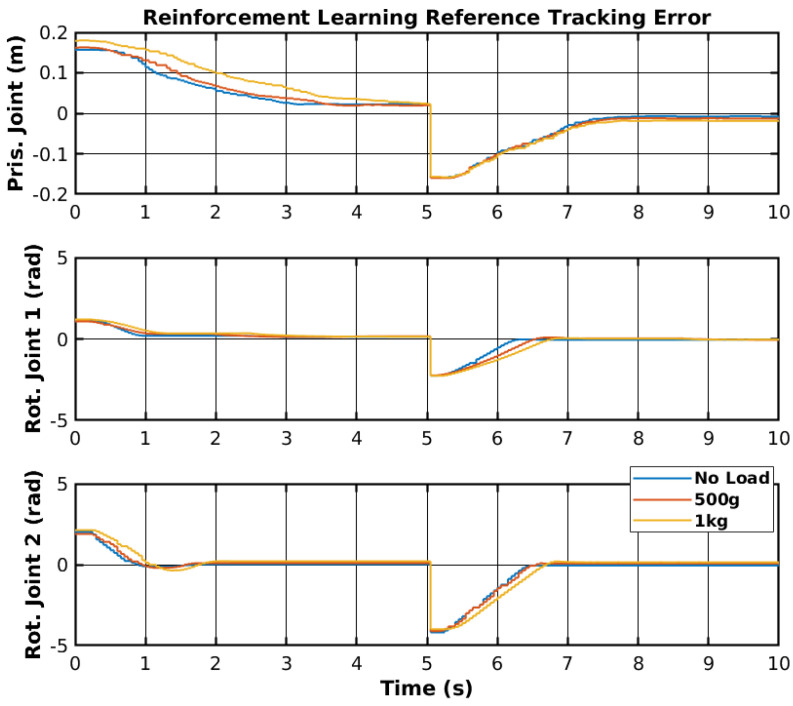
Reinforcement learning reference tracking error.

**Figure 19 sensors-25-02676-f019:**
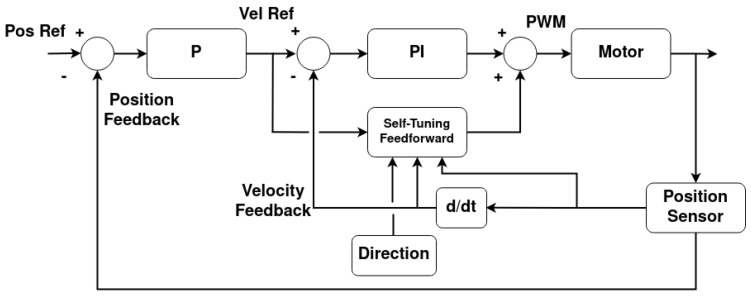
Cascaded P-PI controller with self-tuning feedforward architecture.

**Figure 20 sensors-25-02676-f020:**
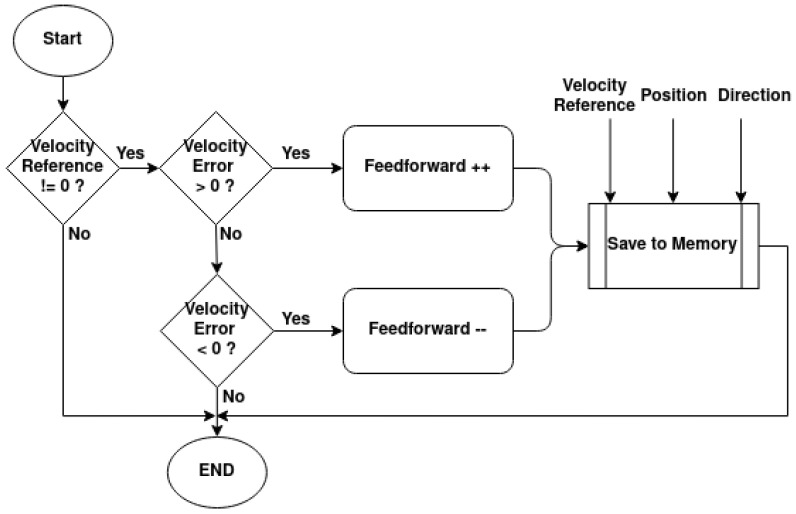
Feedforward value update diagram.

**Figure 21 sensors-25-02676-f021:**
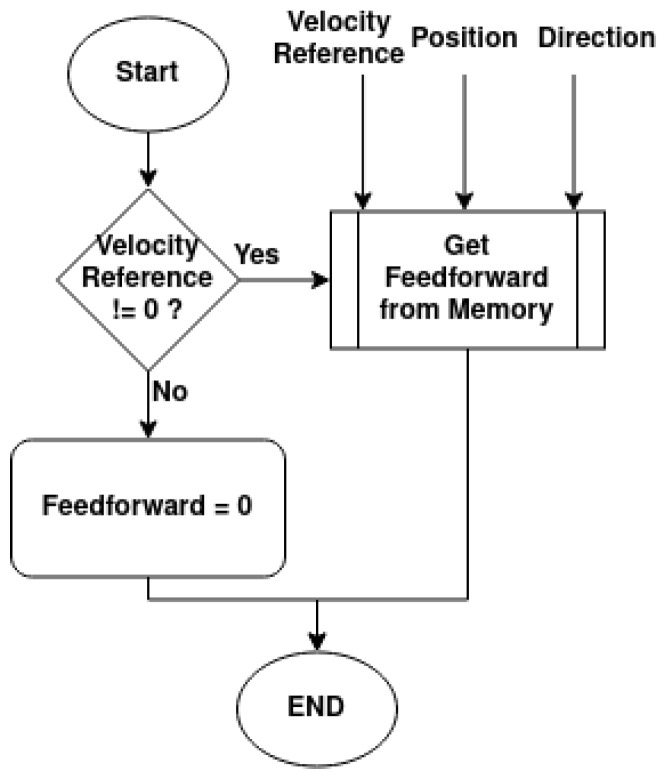
Obtain updated feedforward value diagram.

**Figure 22 sensors-25-02676-f022:**
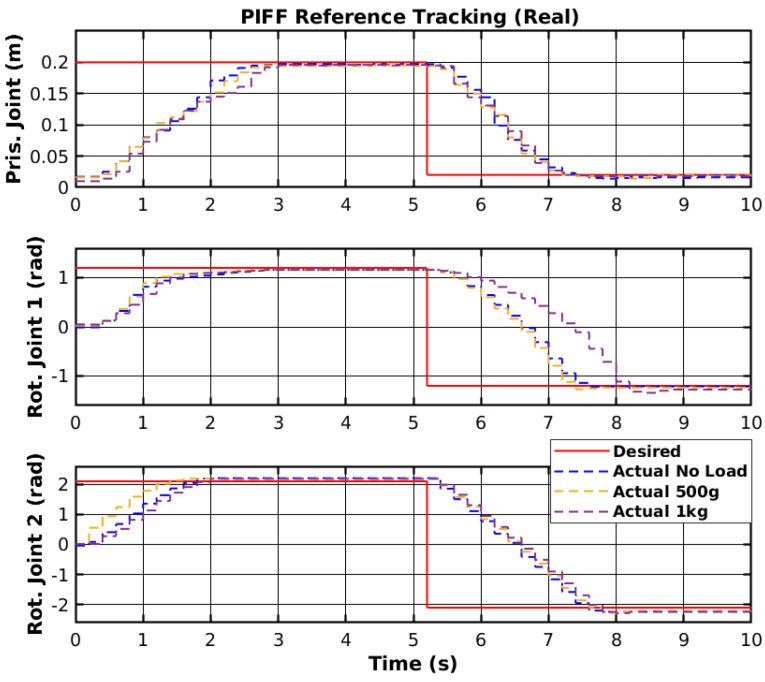
PIFF reference tracking.

**Figure 23 sensors-25-02676-f023:**
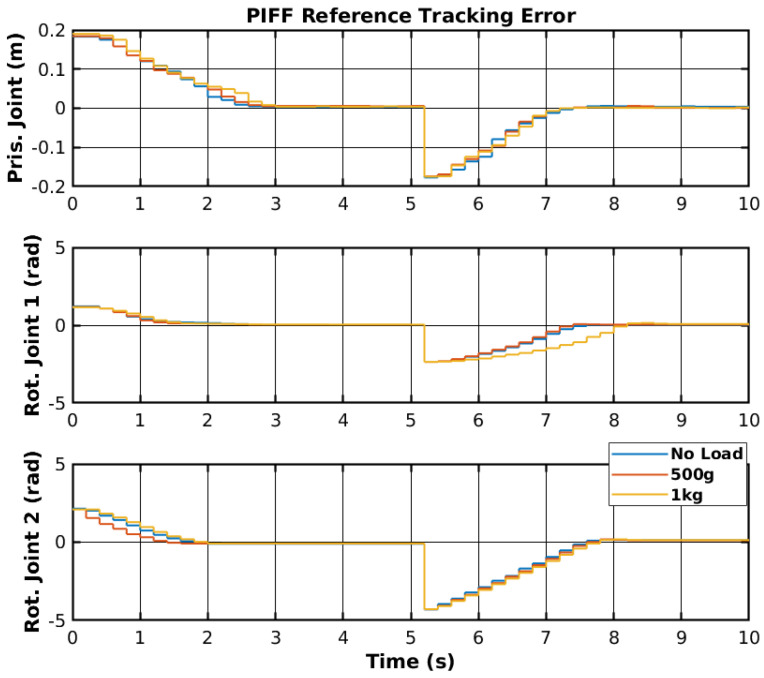
PIFF reference tracking error.

**Table 1 sensors-25-02676-t001:** Denavit–Hartenberg parameters for the manipulator.

Link	$a_{i-1}$	$\alpha_{i-1}$	$d_{i}$	$\theta_{i}$
1	0	0	$d_{1}$	0
2	0	0	0	$\theta_{2}$
3	$l_{2}$	0	0	$\theta_{3}$
4	$l_{3}$	0	0	0

**Table 2 sensors-25-02676-t002:** SMC reference tracking RMS error.

	No Load	500 g	1 kg
**Pris. Joint**	0.092 m	0.0917 m	0.090 m
**Rot. Joint 1**	0.594 rad	0.643 rad	0.670 rad
**Rot. Joint 2**	1.388 rad	1.358 rad	1.447 rad

**Table 3 sensors-25-02676-t003:** DDPG hyperparameter settings.

Hyperparameter	Value
Actor network learning rate	1 × 10^−4^
Critic network learning rate	1 × 10^−4^
Discount factor γ	0.90
Batch size	32
Sample time τ	0.05 s

**Table 4 sensors-25-02676-t004:** RL reference tracking RMS error.

	No Load	500 g	1 kg
**Pris. Joint**	0.075 m	0.079 m	0.091 m
**Rot. Joint 1**	0.604 rad	0.649 rad	0.723 rad
**Rot. Joint 2**	1.098 rad	1.110 rad	1.231 rad

**Table 5 sensors-25-02676-t005:** PIFF reference tracking RMS error.

	No Load	500 g	1 kg
**Pris. Joint**	0.079 m	0.078 m	0.082 m
**Rot. Joint 1**	0.841 rad	0.821 rad	1.000 rad
**Rot. Joint 2**	1.388 rad	1.378 rad	1.465 rad

## Data Availability

The original contributions presented in this study are included in the article. Further inquiries can be directed to the corresponding author.

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
