# Peer review of "Benchmarking Controllers for Low-Cost Agricultural SCARA Manipulators"

_sensors, 2025, doi:10.3390/s25092676_

Round 1
Reviewer 1 Report
Comments and Suggestions for Authors
This work presents the implementation of different control methods in manipulation of low-cost SCARAs. Overall, the paper is well-written with clear explanations of methodologies along with simulation results. However, in terms of originality, it is not clear what technical advantages/disadvantages that the proposed low-cost solution offers in comparison with standard industrial SCARAs. Industrial SCARAs are widely used for various object handling and assembly applications that require high precision. In addition, the reinforcement learning method needs improvements like clarifying the Q term and detailed roles of actors and critics. Line 260 contains an incorrect definition of reinforcement learning. It seems like this paper mostly concerns about robotics and actuation rather than sensing. So, the authors may need to submit this work to journals on robotics or control systems.
Author Response
We would like to thank the reviewer for all the constructive comments, as they have increased the article's quality.
Comment 1: "It is not clear what technical advantages/disadvantages that the proposed low-cost solution offers in comparison with standard industrial SCARAs."
Response 1: Thank you for this comment, we've added the advantages of using low-cost SCARA manipulators to the introduction.
Update in the Document 1: [Lines 41-46] "Compared to traditional industrial SCARA manipulators, which are typically designed for high precision in controlled environments, the proposed low-cost solution targets agricultural applications where flexibility and cost-effectiveness are prioritized over extreme precision, which makes the system more accessible in agriculture where labour shortages and high equipment costs are significant challenges."
Comment 2: "The reinforcement learning method needs improvements like clarifying the Q term and detailed roles of actors and critics.
Response 2: Thank you for pointing this out, we've clarified what the Q term is and the roles of the actor and critic networks.
Update in the Document 2: [Lines 307-311] "the action-value function Q(s,a), shown in Equation 19, estimates the expected cumulative reward achievable from action a in state s. The actor is responsible for selecting actions based on the current state. Simultaneously, the critic evaluates the chosen actions by estimating the value of the action taken and updating the actor's policy accordingly."
Comment 3: "Line 260 contains an incorrect definition of reinforcement learning."
Response 3: Thank you very much for this comment. The definition has been changed.
Update in the Document 3: [Lines 283-290] Reinforcement Learning (RL) is a machine learning paradigm in which an agent learns to make decisions by interacting with an environment and receiving rewards or penalties based on its actions. The agent continuously learns through trial and error, exploring different actions and receiving feedback in the form of rewards, striving to maximise the cumulative reward over time [42,43]. The RL system typically consists of two key components: the actor, which selects actions based on the current policy, and the critic, which evaluates the chosen actions by estimating their value and helping the actor improve its policy. A generic diagram illustrating the RL problem is shown in Figure 14.
Comment 4: It seems like this paper mostly concerns about robotics and actuation rather than sensing. So, the authors may need to submit this work to journals on robotics or control systems.
Response 4: Thank you for this comment, we've submitted to the MDPI Sensors section "Smart Agriculture" because it publishes original peer-reviewed papers on state-of-the-art agriculture robots, and their applications in the field of smart farming.
Reviewer 2 Report
Comments and Suggestions for Authors
This paper proposes a low-cost SCARA manipulator solution for the current situation of labor shortage and introduces three low-cost control strategies, which have certain application value. However, the following several issues need to be addressed.
- When introducing the three controllers, although the basic principles and implementation methods of each controller are described, the basis and optimization process for the selection of controller parameters are not elaborated in detail. It is suggested that in the introduction of each controller, a subsection should be added to specifically discuss the methods and basis for parameter selection, such as determining the switching gain of the sliding mode controller, the reward function parameters of reinforcement learning, and the proportional and integral parameters of the PI controller through stability analysis, trade-offs between response speed and overshoot.
- Although the experimental results section shows the performance of different controllers in simulation and on actual robots, the comparative analysis of the results is not in-depth enough, and there is a lack of detailed discussion on the causes of errors.
- The conclusion part summarizes the three controllers rather briefly and does not fully combine the experimental results to highlight the applicable scenarios and limitations of each controller in the application of agricultural robots.
- There are some places in the text where the language expression is not accurate and smooth, as well as the problem of inconsistent reference formats.
There are some places in the text where the language expression is not accurate and smooth. Please check the paper thoroughly.
Author Response
We would like to thank the reviewer for all the constructive comments and suggestions as they have increased the article's quality.
Comment 1: "When introducing the three controllers, although the basic principles and implementation methods of each controller are described, the basis and optimization process for the selection of controller parameters are not elaborated in detail. It is suggested that in the introduction of each controller, a subsection should be added to specifically discuss the methods and basis for parameter selection, such as determining the switching gain of the sliding mode controller, the reward function parameters of reinforcement learning, and the proportional and integral parameters of the PI controller through stability analysis, trade-offs between response speed and overshoot."
Response 1: Thank you for this comment, the way the sliding mode parameters were obtained was through trial-and-error considering a trade-off between system stability and response time in the simulated environment. In the PI controller, the objective is for it to self adjust using the self-tunning feedforward element, and so PI parameter calibration is not necessary. On the RL controller, the reward function was generated to increase the reward as the error tends to zero.
Update in the document 1: [Lines 238-240] "The switching gain K and the sliding surface gain λ were chosen using trial-and-error and considering the trade-off between system stability and response time in the simulated environment."
Comment 2: "Although the experimental results section shows the performance of different controllers in simulation and on actual robots, the comparative analysis of the results is not in-depth enough, and there is a lack of detailed discussion on the causes of errors."
Response 2: Thank you very much for point this out, a discussion section was added before the conclusion.
Update in the document 2: "The presented controllers could perform trajectory tracking to reach the target reference in a reasonable time. Although the general performance of the controllers was suitable for reference tracking, some issues were present. The sliding mode controller showed the best results; however, the manipulator jittered frequently and did not present smooth joint movement. The reinforcement learning controller did not jitter at all but did not reach the reference on the prismatic joint. Furthermore, the controller performed student stops when reaching the reference at a high velocity, which would cause some overshoots. The PI controller with a self-tuning feedforward element had a smoother response. The problem with this controller is that, given the reduced error as the joint approached the reference, the generated reference velocity would decrease, and the motor would reach the dead zone. The self-tunning feedforward element resolved this issue, but the joint slowed drastically before reaching the reference. Furthermore, the controller is more susceptible to load changes than the previous controllers, as the first rotational joint responded much slower to the 1 kg load."
Comment 3: "The conclusion part summarizes the three controllers rather briefly and does not fully combine the experimental results to highlight the applicable scenarios and limitations of each controller in the application of agricultural robots."
Response 3: Thank you for point this out, the conclusion was updated accordingly.
Update in the document 3: [Lines 454-456] "This aligns with existing literature that acknowledges SMC’s robustness but highlights challenges related to chattering effects [44] which limits its use in tasks requiring fine control and precision, such as delicate harvesting tasks. While it provides quick feedback in dynamic conditions, its lack of smoothness is a disadvantage in tasks where accuracy is critical."
[Lines 461-463] "Furthermore, the computational resources and time required for training can be a barrier, particularly in applications that demand immediate responses to changes, such as fruit harvesting in dynamic environments."
Comment 4: "There are some places in the text where the language expression is not accurate and smooth, as well as the problem of inconsistent reference formats."
Response 4: Thank you for this comment, the document has been revised and corrected.
Reviewer 3 Report
Comments and Suggestions for Authors
This paper compares 3 control strategies for SCARA manipulators in agricultural tasks. Some experiments are carried out to evaluate the performance of the controllers. The Authors should clarify the novelty aspect, since the control techniques are known in literature. The following issues also need to be addressed.
- The proposed manipulator is a serial PRR. Usually, SCARAs perform pick & place tasks with a RRP structure. The Authors should discuss and motivate this design choice, since the first motor of the chain has to compensate for gravity and inertia of the other two links. In this case, the pyramidal effect could be more pronounced.
- The control schemes should be added in the description of each control approach, making clear the controlled variables.
- Some pictures showing the experimental test could help the reader in understanding the task. However, the experimental procedure seems quite general and not specifically applied to the agricultural/harvesting tasks.
- In the introduction, many investigations, also recent, regarding PID control are not considered. The Authors should update their review, including proper references [1-6].
- In the conclusions, disadvantages/limitations are discussed, and it is not clear what the advantages of the proposed control strategies are. In the present form, it seems that the paper find results well-known in literature.
[1] 10.1109/TMECH.2020.3028968
[2] https://doi.org/10.1049/ipr2.12758
[3] https://doi.org/10.1115/1.4067626
[4] 10.1109/LCSYS.2023.3347176
[5] https://doi.org/10.1016/j.aej.2024.05.092
[6] https://doi.org/10.1016/j.oceaneng.2017.04.030
Author Response
We would like to thank the reviewer for all the constructive comments and suggestions, as they have increased the article's quality.
Comment 1: The proposed manipulator is a serial PRR. Usually, SCARAs perform pick & place tasks with a RRP structure. The Authors should discuss and motivate this design choice, since the first motor of the chain has to compensate for gravity and inertia of the other two links. In this case, the pyramidal effect could be more pronounced.
Response 1: Thank you very much for this comment. Using a PRR instead of an RRP, in agriculture, is more beneficial as there isn't much space for a prismatic link near the leaves and branches. With the prismatic link at the back, the end-effector will have more maneuverability near the leaves and branches.
Update in the document 1: [Lines 147-151] "A Proportional-Rotational-Rotational (PRR) configuration was chosen as the workplace (for example, a tomato plant) is confined due to the presence of branches and leaves. Thus, having a smaller area for the tool is beneficial. In an RRP configuration, the area near the tip of the manipulator would be greater, which could compromise the harvesting process."
Comment 2: The control schemes should be added in the description of each control approach, making clear the controlled variables.
Response 2: Thank you for this comment, the control variables have been detailed before each equation as there were instances where this was not happening.
Comment 3: Some pictures showing the experimental test could help the reader in understanding the task. However, the experimental procedure seems quite general and not specifically applied to the agricultural/harvesting tasks.
Response 3: Thank you for this comment. The purpose is to develop and benchmark controllers that can be used in an agricultural environment using a low-cost manipulator. Although the experiments were not done in an agricultural environment, if the low-cost manipulator with high uncertainties can be controlled adequately, then a low-cost alternative for farmers interested in using robotics is possible.
Comment 4: In the introduction, many investigations, also recent, regarding PID control are not considered. The Authors should update their review, including proper references [1-6].
Response 4: Thank you for pointing this out, we have added two more references in the related work section. However, since this article isn't focused on generic PID control, we did not go into as much detail as we did with sliding mode and reinforcement learning.
Update in the document 4: [Lines 65-69] "These classical controllers are used in a wide range of manipulator applications and are often combined with other control methodologies. For example, Shojaei et al. [14] developed an observer-based neural adaptive PID controller for robotic manipulators and Londhe et al. [15] developed a PID fuzzy control scheme for underwater vehicle-manipulators."
Comment 5: In the conclusions, disadvantages/limitations are discussed, and it is not clear what the advantages of the proposed control strategies are. In the present form, it seems that the paper find results well-known in literature.
Response 5: Thank you for pointing this out. We have added more details regarding this comment in the conclusions and also added a discussion section.
Update in the document 5: [Lines 428-441]: "The presented controllers could perform trajectory tracking to reach the target reference in a reasonable time. Although the general performance of the controllers was suitable for reference tracking, some issues were present. The sliding mode controller showed the best results; however, the manipulator jittered frequently and did not present smooth joint movement. The reinforcement learning controller did not jitter at all but did not reach the reference on the prismatic joint. Furthermore, the controller performed student stops when reaching the reference at a high velocity, which would cause some overshoots. The PI controller with a self-tuning feedforward element had a smoother response. The problem with this controller is that, given the reduced error as the joint approached the reference, the generated reference velocity would decrease, and the motor would reach the dead zone. The self-tunning feedforward element resolved this issue, but the joint slowed drastically before reaching the reference. Furthermore, the controller is more susceptible to load changes than the previous controllers, as the first rotational joint responded much slower to the 1 kg load."
[Lines 454-456] : "This aligns with existing literature that acknowledges SMC’s robustness but highlights challenges related to chattering effects [44] which limits its use in tasks requiring fine control and precision, such as delicate harvesting tasks. While it provides quick feedback in dynamic conditions, its lack of smoothness is a disadvantage in tasks where accuracy is critical."
[Lines 461-463] : "Furthermore, the computational resources and time required for training can be a barrier, particularly in applications that demand immediate responses to changes, such as fruit harvesting in dynamic environments."
Finally, the document was revised to correct and improve the english.
Round 2
Reviewer 1 Report
Comments and Suggestions for Authors
Thank you for revising. Figure 14 and Figure 15 show redundant diagrams. Figure 15 is more relatable than Figure 14. So, there is no need to keep Figure 14.
Author Response
Thank you very much.
I've deleted Figure 14 and altered Figure 15 to have a dotted square around the actor and critic nodes and called it agent.
Reviewer 2 Report
Comments and Suggestions for Authors
it is ok for me.
Author Response
Thank you very much.
Reviewer 3 Report
Comments and Suggestions for Authors
The Authors failed in answering to question 1. The question is very specific about the kinematics and the pyramidal effect, and the answer regarding the "area" is not clear and ambiguous. What area? The workspace region is a volume, and it does not vary with the order of the joints. I do not agree with answer 4. In the review in the introduction, many relevant works should be aknowledge.
Author Response
Thank you very much for your comments.
My apologies for not answering your question.
Considering the first question. I have revised the text and added more context, corrected the area to volume and referred to the pyramidal effect.
Here is the corrected text:
[Lines 153-165] "A Proportional-Rotational-Rotational (PRR) configuration was selected due to the confined nature of the workspace (for example, a tomato plant), where dense foliage, including branches and leaves, limits available space. In such conditions, a smaller tool volume is advantageous, as it improves manoeuvrability and reduces the risk of damaging the plant. In contrast, an RRP configuration typically results in a larger volume near the manipulator’s tip, which may hinder operation in these restricted environments. Additionally, employing a prismatic joint to lift the manipulator allows the actuator to be positioned at the base, shifting the centre of mass closer to the robot’s base. This reduces the torque required by the rotational joints, although it increases the force demand on the prismatic actuator. Despite the simplicity of this kinematic structure, it is subject to the pyramidal effect, where small angular errors and velocities in the proximal joints become amplified as linear errors and velocities at the end-effector. "
For question 4, I have added more literature examples of PID in manipulators, specifically combinations of PID with other controllers.
Here are the added examples:
[Lines 60 - 74] "Classical robotic manipulator control relies on well-established PID-based and model-based controllers. PID controllers are widely used due to their simplicity and effectiveness in position control [12]. However, they struggle with nonlinearities, disturbances, and unmodeled dynamics, requiring additional compensators. Model Predictive Control (MPC) is another approach that optimises control inputs over a finite horizon, allowing constraint handling and smoother trajectory tracking [13].
These classical controllers are used in a wide range of manipulator applications and are often combined with other control methodologies. For example, Shojaei et al. [14] developed an observer-based neural adaptive PID controller for robotic manipulators, Londhe et al. [15] developed a PID fuzzy control scheme for underwater vehicle-manipulators, Heidar et al. [16] proposed a PID fuzzy controller for a parallel manipulator. Kumar et al. [17] developed a neural network-based PID controller for a one-link manipulator. This controller allowed the use of an unknown system model and could identify system uncertainties that prevailed during the manipulator's operation. Tang et al. [18] proposed a self-adaptive PID controller based on a radial basis function neural network to solve the weak adaptive ability and poor robustness of a conventional PID controller."